# CheXseen: Unseen Disease Detection for Deep Learning Interpretation of Chest X-rays

**Siyu Shi**[*]                                                                    SIYUSHI@STANFORD.EDU

**Ishaan Malhi**[*]                                                               IMALHI@CS.STANFORD.EDU

**Kevin Tran**                                                                  KTRAN23@STANFORD.EDU

**Andrew Y. Ng**                                                                   ANG@CS.STANFORD.EDU

**Pranav Rajpurkar**                                                      PRANAVSR@CS.STANFORD.EDU

*Stanford University*

## Abstract

We systematically evaluate the performance of deep learning models in the presence of diseases not labeled for or present during training. First, we evaluate whether deep learning models trained on a subset of diseases (seen diseases) can detect the presence of any one of a larger set of diseases. We find that models tend to falsely classify diseases outside of the subset (unseen diseases) as "no disease". Second, we evaluate whether models trained on seen diseases can detect seen diseases when co-occurring with diseases outside the subset (unseen diseases). We find that models are still able to detect seen diseases even when co-occurring with unseen diseases. Third, we evaluate whether feature representations learned by models may be used to detect the presence of unseen diseases given a small labeled set of unseen diseases. We find that the penultimate layer of the deep neural network provides useful features for unseen disease detection. Our results can inform the safe clinical deployment of deep learning models trained on a non-exhaustive set of disease classes.

**Keywords:** Unseen Disease Detection, Deep Learning

## 1. Introduction

Safe clinical deployment of deep learning models for disease diagnosis would require models to not only diagnose diseases that they have been trained to detect, but also recognize the presence of diseases they have not been trained to detect for possible deferral to a human expert (Mozannar and Sontag, 2021; Rajpurkar et al., 2020). Medical imaging datasets used to train models typically only provide labels for a limited number of common diseases because of the challenge and costs associated with labeling for all possible diseases. For example, some serious diseases, including pneumomediastinum, are not part of any commonly used chest X-ray databases (Irvin et al., 2019; Johnson et al., 2019; Wang et al., 2017). However, it is unknown whether deep learning models for chest x-ray interpretation can maintain performance in presence of diseases not seen during training, or whether they can detect the presence of such diseases.

In this study, we provide a systematic evaluation of deep learning models in the presence of diseases not labeled for or present during training. Specifically, we first evaluate whether deep learning models trained on a subset of diseases (seen diseases) can detect the presence

---

[*] Contributed equally

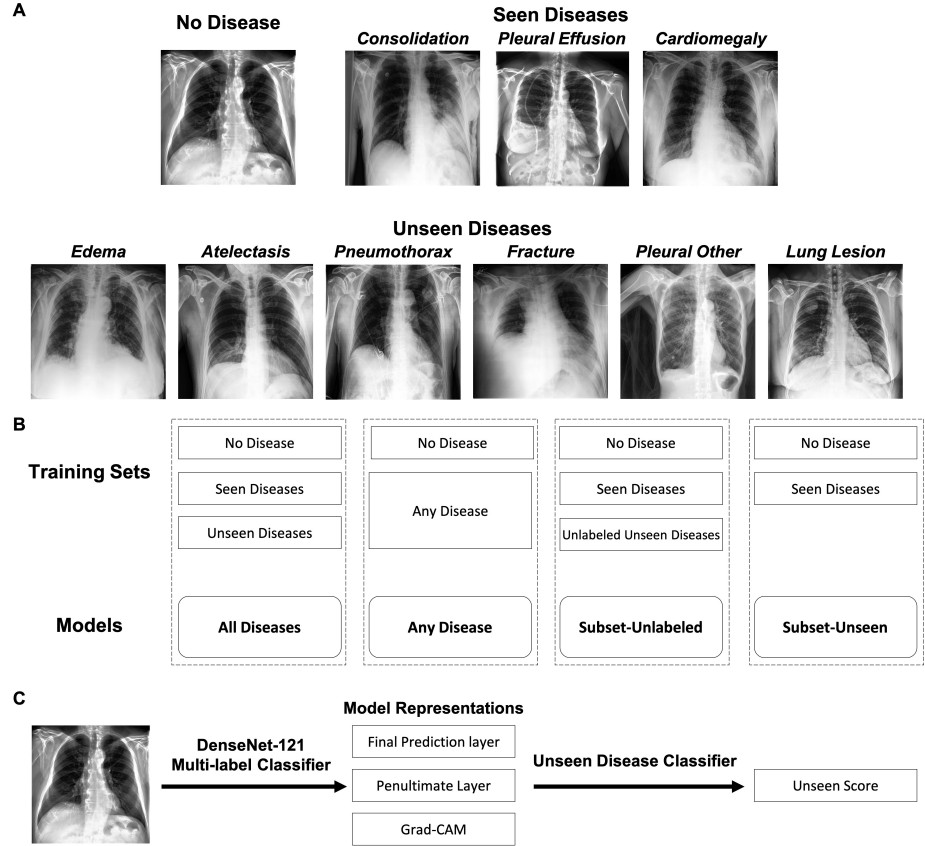

Figure 1: Overview of the experimental setup. A. Chest X-ray image labels include "No Disease", three seen diseases, and six unseen diseases. B. Training data setup for the four multi-label models. The Subset-Unlabeled model is trained with all images in the train set, with the labels of "No Disease" and three seen diseases, while excluding the labels of six unseen diseases. The Subset-Unseen model is trained with only the images that do not have any of the six unseen diseases. The All Diseases model is trained with all images and all ten labels ("seen" diseases, "unseen" diseases and "no disease"), and serves as a control. The Any Disease model is trained with all images for either having "any disease" or "no disease", and serves as another control. C. Three outputs from the models - final prediction layer, penultimate (intermediate) layer, and visualization map are used to train unseen disease classifiers, to predict the "unseen score" (whether an unseen disease is present during testing).

of any one of a larger set of diseases. We find that models tend to falsely classify diseases outside of the subset (unseen diseases) as "no disease". Second, we evaluate whether models trained on seen diseases can detect seen diseases when co-occurring with diseases outside the subset (unseen diseases). We find that models are still able to detect seen diseases even

when co-occurring with unseen diseases. Third, we conduct an initial exploration of unseen disease detection methods, focused on evaluation of feature representations. We evaluate whether feature representations learned by models may be used to detect the presence of unseen diseases given a small labeled set of unseen diseases. We find that the penultimate layer provides useful features for unseen disease detection. Our results can inform the safe clinical deployment of deep learning models trained on a non-exhaustive set of disease classes.

## 2. Related Work

Traditional machine learning frameworks assume a "closed world" assumption, where no new classes exist in the test set. However, in real world applications, trained models could encounter new classes. Deep learning models for image recognition are known to suffer in performance when applied to a test distribution that differs from their training distribution (Hendrycks and Gimpel, 2016; Quionero-Candela et al., 2009; Sathitratanacheewin and Pongpirul, 2018; Pooch et al., 2019). Several methodologies have been explored in computer vision for novelty or abnormality detection, including reconstruction-based methods, self-representation, statistical modeling, and deep adversarial learning (Pimentel et al., 2014; Xu et al., 2015; Markou and Singh, 2003). Several methodologies were explored for open set clinical decision making showing no clear superior techniques (Kingma et al., 2019). However, most methodologies are designed for multiclass problems but not multi-label problems (Geng et al., 2020; Bendale and Boult, 2016). Specifically for healthcare applications, there have been limited studies on out-of-distribution medical imaging (Cao et al., 2020; Mårtensson et al., 2020), with no previous studies investigating the performance of medical imaging classification models when facing unseen diseases.

## 3. Methods

### 3.1. Data

We form a dataset which has disease labels split into two categories: "seen diseases" and "unseen diseases" (Figure 1 A). We modify the CheXpert dataset, consisting of 224,316 chest radiographs from 65,240 patients labeled for the presence of 14 observations (Irvin et al., 2019). To be able to split disease labels without overlap, we remove children and parent label classes shared by diseases, specifically the Enlarged Cardiomediastinum, Airspace Opacity and Pneumonia label classes. We also remove the Support Devices label, as it is a clinically insignificant observation. We divide the remaining labels into four seen labels (No Disease, Consolidation, Pleural Effusion, Cardiomegaly), and six unseen labels (Pleural Other, Edema, Lung Lesion, Atelectasis, Fracture, and Pneumothorax). This division is based on each disease's prevalence in the dataset in order to evaluate model performance when trained only on commonly occurring diseases (Figure 1A). We use the CheXpert validation set to select models and to train unseen disease classifiers, which is a set of 200 labeled studies, where ground truth was set by annotation from a consensus of 3 radiologists. We use the CheXpert test set, which consists of 500 chest x-ray studies annotated with a radiologist majority vote, to evaluate the performance of models.

### 3.2. Multi-Label Models

We train four multi-label models with different sets of images and labels, and evaluate the multi-label models on their disease detection performance. An overview of the models' setup is outlined in Figure 1 B.

**All Diseases (Control)**  The All Diseases model is trained with all images and all ten disease labels ("no disease" and both seen and unseen), and serves as a comparison to models trained on a subset of diseases.

**Any Disease (Control)**  The Any Disease model is trained with all images as a binary model with the "no disease" label (signifying a normal chest X-ray without any disease) and serves as another control comparison.

**Subset-Unlabeled**  The Subset-Unlabeled model is trained with all images, but with the labels for unseen diseases removed. In the Subset-Unlabeled model, all image studies are included in training, while removing the six unseen disease labels.

**Subset-Unseen**  The Subset-Unseen model is trained with images that have either no disease or have only seen diseases. Image studies with one or more unseen labels are removed, while also removing the six unseen disease labels.

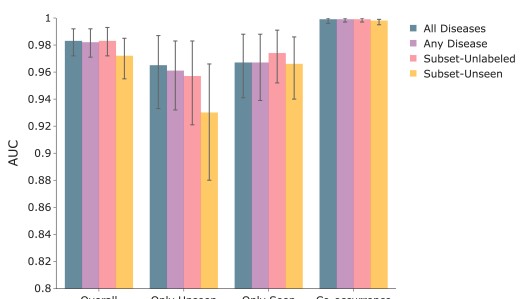

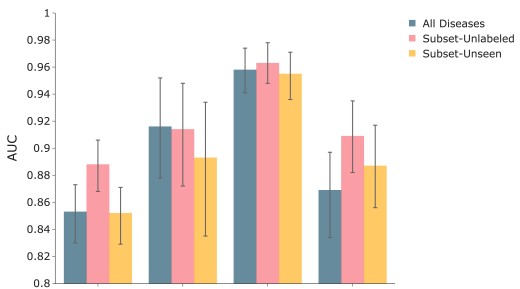

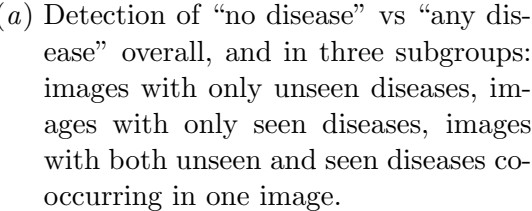

($a$) Detection of "no disease" vs "any disease" overall, and in three subgroups: images with only unseen diseases, images with only seen diseases, images with both unseen and seen diseases co-occurring in one image.

($b$) Performance of models in detecting seen diseases overall and for individual seen diseases: consolidation, pleural effusion and cardiomegaly. The overall evaluation strategy considers the average AUC over all seen diseases.

Figure 2: Performance of multi-label models under various setups.

## 4. Statistical analysis

To determine statistical significance between 2 models, we use the 95% confidence intervals of the difference between bootstrap samples. To generate confidence intervals, we used the non-parametric bootstrap with 1000 bootstrap replicates. Statistically significant differences between models were established using the non-parametric bootstrap on the mean AUC

difference on the test set. We calculate p-values from the confidence interval using the method described in (Altman and Bland, 2011) with a threshold of 0.05 for hypothesis testing. This method was chosen to evaluate whether 2 models were similar in performance with respect to their average AUC over the bootstrap sample, and to test statistically significant performance differences in either direction using the 95% confidence intervals. We use the Benjamini-Hochberg method to correct for multiple hypothesis testing between various models.

## 5. Detection of any disease vs no disease

We evaluate the performance of the multi-label models on detecting the presence of any disease (vs "no disease") on a test set containing both seen and unseen diseases. Results are summarized in Figure 2(a), and Tables 1 and 2.

**Subset-Unlabeled vs Controls**   The Subset-Unlabeled model is not statistically significantly different from the Any Disease model (mean AUC difference 0.001, [95% CI -0.004, 0.005]), and the All Diseases model (mean AUC difference 0.000, [95% CI -0.003, 0.003]).

**Subset-Unseen vs Controls**   The Subset-Unseen model performs statistically significantly lower than the Any Disease model overall (mean AUC difference -0.010, [95% CI -0.019,-0.004]), but is not statistically significantly different to the Any Disease model when evaluating examples with only seen diseases (mean AUC difference -0.002, [95% CI -0.015,0.015]). We find that the Subset-Unseen model performs statistically significantly lower than the All Diseases model overall (mean AUC difference -0.010, [95% CI -0.018, -0.003]), but is not statistically significantly different in evaluating examples with co-occurring seen and unseen diseases (mean AUC difference -0.004, [95% CI -0.010, 0.000]).

**Subset-Unlabeled vs Subset-Unseen**   The Subset-Unlabeled model performs statistically significantly higher than the Subset-Unseen model in detecting "no disease" vs "any disease" in the presence of only unseen diseases (mean AUC difference 0.028 , [95% CI 0.011, 0.047]), and in the presence of co-occurring seen and unseen diseases (mean AUC difference 0.004, [95% CI 0.001, 0.009]). The Subset-Unlabeled model is not statistically significantly different from the Subset-Unseen model for only seen diseases (mean AUC difference -0.008, [95% CI -0.019, 0.001]).

## 6. Detection of seen diseases in the presence of seen and unseen diseases

We evaluate whether a multi-label model trained on seen diseases can successfully detect seen diseases on a test set containing both seen and unseen diseases. Results are summarized in Figure 2(b), Table. 3 and 4.

**Subset-Unseen vs All Diseases**   The Subset-Unseen model is not statistically significantly different from the All Diseases model overall in detecting seen diseases (mean AUC difference -0.011, [95% CI -0.020, 0.000]).

**Subset-Unlabeled vs All Diseases**   The Subset-Unlabeled model has statistically significantly higher performance when compared to the All Diseases model in detecting seen

diseases overall (mean AUC difference 0.033, [95% 0.025, 0.042]) and in detecting cardiomegaly (mean AUC difference 0.032, [95% CI 0.019, 0.046]).

**Subset-Unseen vs Subset-Unlabeled** The Subset-Unseen model has a statistically significantly lower performance when compared to the Subset-Unlabeled model in detecting seen diseases overall (mean AUC difference -0.044, [95% CI -0.054, -0.033]), pleural effusion (mean AUC difference -0.014, [95% CI -0.025, -0.004]), and cardiomegaly (mean AUC difference -0.019, [95% CI -0.033, -0.005]), and is not statistically significantly different from the Subset-Unlabeled model in detecting consolidation (mean AUC difference -0.023, [95% CI -0.067, 0.020]).

## 7. Unseen disease detection

We develop classifiers to detect the presence of any unseen disease given an X-ray image. Applying the four trained multi-label models to the validation set, we collect the outputs from the final prediction layer, the penultimate layer, and the visualization map (generated using the Grad-CAM method (Selvaraju et al., 2017)). The output of the visualization map using GradCAM is used as a matrix directly in the following steps. The feature representations are extracted from running the validation set on the trained classification models. The three sets of outputs are then used to train a random forest classifier and a logistic regression classifier, with a binary outcome on whether the chest X-ray has an unseen disease or not, to produce an "unseen score" using unseen disease labels on the validation set (shown in Figure 1C). Logistic regression classifier is a commonly used standard for binary classification. Random forest classifiers, compared to logistic regression, are able to create nonlinear decision boundaries. Unseen scores are the output of the random forest classifier or the logistic regression classifier, and a numeric number between 0 and 1, signifying how likely the chest X-ray image has an unseen disease. The performance of these classifiers is reported on the test set. Results are summarized in Figure 3 and Table 5.

**Comparing feature representations** Unseen scores derived from the penultimate layer have the best average performance (AUC 0.873, [95% CI 0.848, 0.897]), followed by those from the final prediction layer (AUC 0.860, [95% CI 0.833, 0.889]) and the visualization map (AUC 0.851 [95% CI 0.832, 0.879]). The performance of unseen scores derived from the penultimate layer is statistically significantly higher than those from the final prediction layer (mean AUC difference 0.013 [95% CI 0.009, 0.017]), which is higher than those from the visualization map (mean AUC difference 0.009 [95% CI 0.007, 0.011]).

**Comparing classifiers** Over all of the representations and the multi-label models, the random forest classifier has a high average performance of AUC 0.862 [95% CI 0.837, 0.892], but this is not statistically significantly higher than the performance of the logistic regression classifier (mean AUC difference 0.002 [95% CI 0.000, 0.003]).

**Comparing multi-label models** Using the random forest classifier trained on the penultimate layer from the four multi-label models, the unseen score derived from the Any Disease model has the best performance (mean AUC 0.879, [95% CI 0.849, 0.901]) at predicting the presence of unseen disease, followed by the unseen score from the All Diseases model (mean AUC 0.875, [95% CI 0.850, 0.899]). The performance of the unseen score from the Any

Diseases model is statistically significantly higher than that of the unseen score derived from the All Diseases model (mean AUC difference 0.004, [95% CI 0.003, 0.006]). The unseen scores derived from the Subset-Unlabeled and the Subset-Unseen models have the lowest performance among the unseen scores of the four models (AUC 0.874, [95% CI 0.846, 0.897]) and (AUC 0.870, [95% CI 0.842, 0.894]) respectively. Finally, the performance of the unseen scores derived from the Subset-Unlabeled model is statistically significantly higher than that of the Subset-Unseen model (mean AUC difference 0.005, [95% CI 0.003, 0.007]).

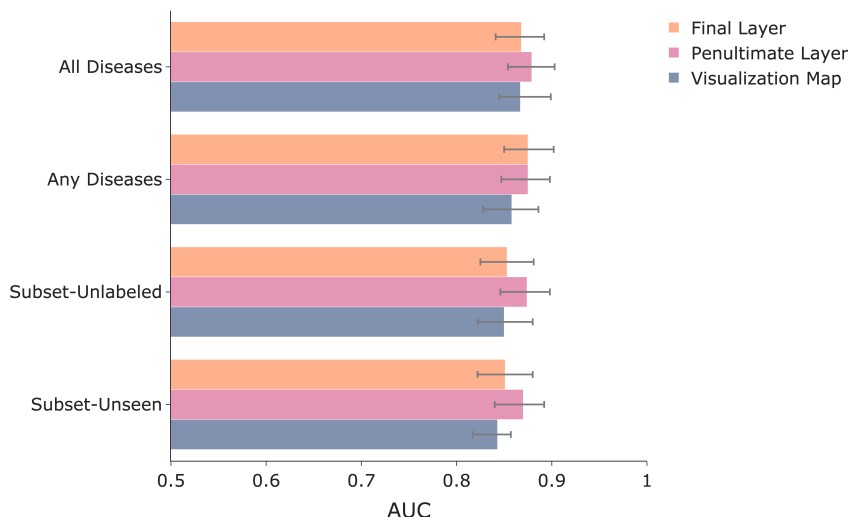

Figure 3: Performance on the task of unseen disease detection using unseen scores. Unseen scores were outputted by random forest classifiers trained using three different feature representations to detect the presence of unseen disease(s): the final prediction layer, penultimate layer and visualization map of the trained multi-label classifiers.

## 8. Limitations

There are three main limitations of our study. First, the dataset has a limited number of diseases, with six unseen diseases in this study. Ideally, a wider variety of unseen diseases would be evaluated to minimize the impact of disease correlations on performance evaluation. Moreover, the ability to expand towards more diseases while maintaining performance is important for a useful model in the actual clinical setting. Second, our study is limited to an internal validation set without an external test set including different unseen diseases. Third, our study did not explore training strategies for multi-label models that could mitigate the performance drop with the All Diseases model compared to the Seen-Unlabeled model.

## 9. Discussion

In this study, we evaluate the performance of deep learning models in the presence of diseases not labeled for or present during training.

**Can models detect seen diseases in the presence of unseen diseases?** Our results show that the Subset-Unlabeled model, which is trained with unseen disease examples but not unseen disease labels, and the Subset-Unseen model, trained without unseen disease examples or labels, are able to detect "any disease" vs "no disease" in images with co-occurring unseen and seen diseases. However, their performance decreases when facing images with only unseen diseases (Figure 2(*a*)). These results show that in a real-world clinical setting, deep learning models may succeed in identifying "no disease" vs "any disease" when an unseen disease co-occurs with a seen disease, but may likely falsely report "no disease" if an unseen disease appears alone. Such mistake can result in delays in correct diagnosis and treatment, and therefore can be life-threatening in some medical conditions (Baiu and Spain, 2019). This result re-emphasizes the necessity for unseen disease detection to avoid misclassification of unseen diseases as "no disease."

Our results also show that the Subset-Unlabeled model and the Subset-Unseen model are able to detect seen diseases, even in the presence of unseen diseases, at a level comparable to the All Diseases model. We find that the Subset-Unlabeled model performs better than the All Diseases model, which may be because of the multi-task nature of the problem, where the optimization landscape may cause detrimental gradient interference between the different tasks and impede learning (Yu et al., 2020). We find that the Subset-Unlabeled model has a statistically significantly higher performance compared to the Subset-Unseen model, likely because the Subset-Unlabeled model is exposed to additional training examples.

**Can unseen diseases be detected without explicit training?** On unseen disease detection, we conduct an initial exploration of unseen disease detection methods, borrowing philosophy from few shot learning, while focusing on evaluating feature representations extracted from classification models. We find that the unseen scores from the Subset-Unlabeled model has higher performance than those from the Subset-Unseen model, likely because the Subset-Unlabeled model learns representations of the unlabeled diseases during training. We find that unseen scores from the penultimate layer are the best for unseen disease detection, followed by the final layer and the visualization map. A possible explanation is that the penultimate layer contains information representing the unseen diseases, whereas the final prediction layer discards this information to reduce training loss. We find that the visualization map is outperformed by both the penultimate and the final prediction layer, perhaps because some diseases in our dataset, including lung lesion, pneumothorax, fracture, atelectasis, can occur in different locations in the chest X-ray than the seen diseases. Overall, our results demonstrate that using feature representations of multi-label models trained on diseases form suitable baselines for unseen disease detection. Exploration of the optimal model for training the unseen disease classifiers using the feature representations evaluated in this work would be an important future research direction.

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

## Appendix A. Supporting Tables

| | Overall | Only Unseen Diseases | Only Seen Diseases | Co-occurring Seen and Unseen Diseases |
|---|---|---|---|---|
| All Diseases | 0.980 (0.966,0.990) | 0.957 (0.922,0.982) | 0.962 (0.932,0.984) | 0.999 (0.996,1.000) |
| Any Disease | 0.982 (0.971,0.992) | 0.961 (0.932,0.983) | 0.967 (0.939,0.988) | 0.999 (0.997,1.000) |
| Subset-Unlabeled | 0.983 (0.972,0.993) | 0.957 (0.921,0.983) | 0.974 (0.952,0.991) | 0.999 (0.997,1.000) |
| Subset-Unseen | 0.975 (0.959,0.988) | 0.937 (0.891,0.971) | 0.968 (0.941,0.987) | 0.997 (0.991,0.999) |

Table 1: Performance in detecting "no disease" vs "any disease" overall and by each subgroup [mean area under curve (AUC), (95% confidence interval)].

| | Overall | Only Unseen Diseases | Only Seen Diseases | Co-occurring Seen and Unseen Diseases |
|---|---|---|---|---|
| Any Diseases | -0.001 (-0.007,0.004) [p: 0.791] | -0.004 (-0.018,0.007) [p: 0.583] | 0.000 (-0.011,0.010) [p: 1.0] | 0.000 (-0.001,0.002) [p: 1.0] |
| Subset-Unlabeled | 0.000 (-0.003, 0.003) [p: 1.000] | -0.007 (-0.017,0.000) [p: 0.124] | 0.006 (-0.006,0.017) [p: 0.396] | 0.000 (-0.001,0.001) [p: 1.000] |
| Subset-Unseen | -0.010 (-0.018, -0.003) [p: 0.006] | -0.035 (-0.059,-0.016) [p: 0.002] | -0.001 (-0.015,0.012) [p: 0.961] | -0.004 (-0.010,0.000) [p: 0.149] |

Table 2: Differences in performance in detecting "no disease" vs "any disease" overall and by each subgroup, compared to the All Diseases model [mean area under curve (AUC), (95% confidence interval)] and p-values with $\alpha \leq 0.05$.

| | Overall | Consolidation | Pleural Effusion | Cardiomegaly |
|---|---|---|---|---|
| All Diseases | 0.851 (0.828,0.871) | 0.910 (0.870,0.948) | 0.956 (0.938,0.971) | 0.863 (0.829,0.894) |
| Subset-Unlabeled | 0.888 (0.868,0.906) | 0.914 (0.872,0.948) | 0.963 (0.948,0.978) | 0.909 (0.882,0.935) |
| Subset-Unseen | 0.861 (0.839,0.881) | 0.909 (0.863,0.947) | 0.958 (0.942,0.974) | 0.886 (0.856,0.913) |

Table 3: Performance in detecting seen diseases overall and by each disease [mean area under curve (AUC), (95% confidence interval)].

| | Overall | Consolidation | Pleural Effusion | Cardiomegaly |
|---|---|---|---|---|
| Subset-Unlabeled | 0.033 (0.025, 0.042) [p: 1.662e-13] | -0.003 (-0.030,0.022) [p: 0.776] | 0.005 (-0.003, 0.015) [p: 0.307] | 0.032 (0.019, 0.046) [p: 2.520e-06] |
| Subset-Unseen | -0.011 (-0.020, 0.000) [p: 0.839] | -0.027 (-0.069,0.021) [p: 0.352] | -0.009 (-0.019, 0.002) [p: 0.104] | 0.012 (-0.001, 0.025) [p: 0.076] |

Table 4: Differences in performance in detecting "no disease" vs "any disease" overall and by each subgroup, compared to the All Diseases model [mean area under curve (AUC), (95% confidence interval)] and p-values with $\alpha \leq 0.05$.

|  | All Diseases | Any Disease | Subset-Unlabeled | Subset-Unseen |
|---|---|---|---|---|
| **Final Prediction layer** | | | | |
| Logistic Regression | 0.863 (0.841,0.898) | 0.871 (0.845,0.897) | 0.850 (0.822,0.878) | 0.848 (0.822,0.877) |
| Random Forest | 0.868 (0.839,0.897) | 0.875 (0.848,0.899) | 0.853 (0.828,0.880) | 0.851 (0.823,0.879) |
| **Penultimate Layer** | | | | |
| Logistic Regression | 0.874 (0.848,0.899) | 0.875 (0.847,0.899) | 0.872 (0.845,0.898) | 0.870 (0.843,0.895) |
| Random Forest | 0.875 (0.850,0.899) | 0.879 (0.849,0.901) | 0.874 (0.846,0.897) | 0.870 (0.842,0.894) |
| **Visualization Map** | | | | |
| Logistic Regression | 0.850 (0.823,0.879) | 0.856 (0.828,0.882) | 0.844 (0.816,0.871) | 0.843 (0.813,0.871) |
| Random Forest | 0.858 (0.826,0.883) | 0.867 (0.841,0.873) | 0.850 (0.820,0.878) | 0.843 (0.815,0.873) |

Table 5: Performance in detecting unseen diseases [mean area under curve (AUC), (95% confidence interval)]. We used three different representations to predict the presence of unseen disease(s): the final prediction layers, penultimate layers and visualization maps from the trained classifiers. For each representation, we trained a logistic regression model and a random forest model.

## Appendix B. Supplementary Information

### B.1. Model Training

During training, the uncertain label is treated as a different class, resulting in a 3-class classifier for each label (negative, uncertain, positive). We use DenseNet121 for all experiments (Huang et al., 2017). Images are fed into the network with size $320 \times 320$ pixels. We use the Adam optimizer with default $\beta$-parameters of $\beta_1 = 0.9$, $\beta_2 = 0.999$ and a fixed learning rate $1 \times 10^{-4}$ (Kingma and Ba, 2014). Batches are sampled using a fixed batch size of 16 images. We train for 3 epochs, saving checkpoints every 4800 iterations. Model training and inference code is written using PyTorch 1.5 on a Python 3.6.5 environment and run on 2 Nvidia GTX 1070 GPUs.

### B.2. Forming Ensembles for Evaluation

For evaluating the performance of the multi-label models, we formed an ensemble of each model by running the model three times with different random initializations. Each run produced 10 top checkpoints. We created an ensemble of the 30 generated checkpoints on the validation set by computing the mean of the output probabilities over the 30 checkpoints for each task.

### B.3. Visualizations of feature representations

To demonstrate the effectiveness of feature representations of multi-label models as inputs to unseen classifiers, we plot the 2D t-SNE (Van der Maaten and Hinton, 2008) clusters in Figure 4 of the feature representations for each multi-label model. The t-SNE was run using a perplexity of 30 with 1000 iterations and a learning rate of 200. The clusters are color coded with the disease subset label from the validation set (seen/unseen) that we describe in Section 3.1.

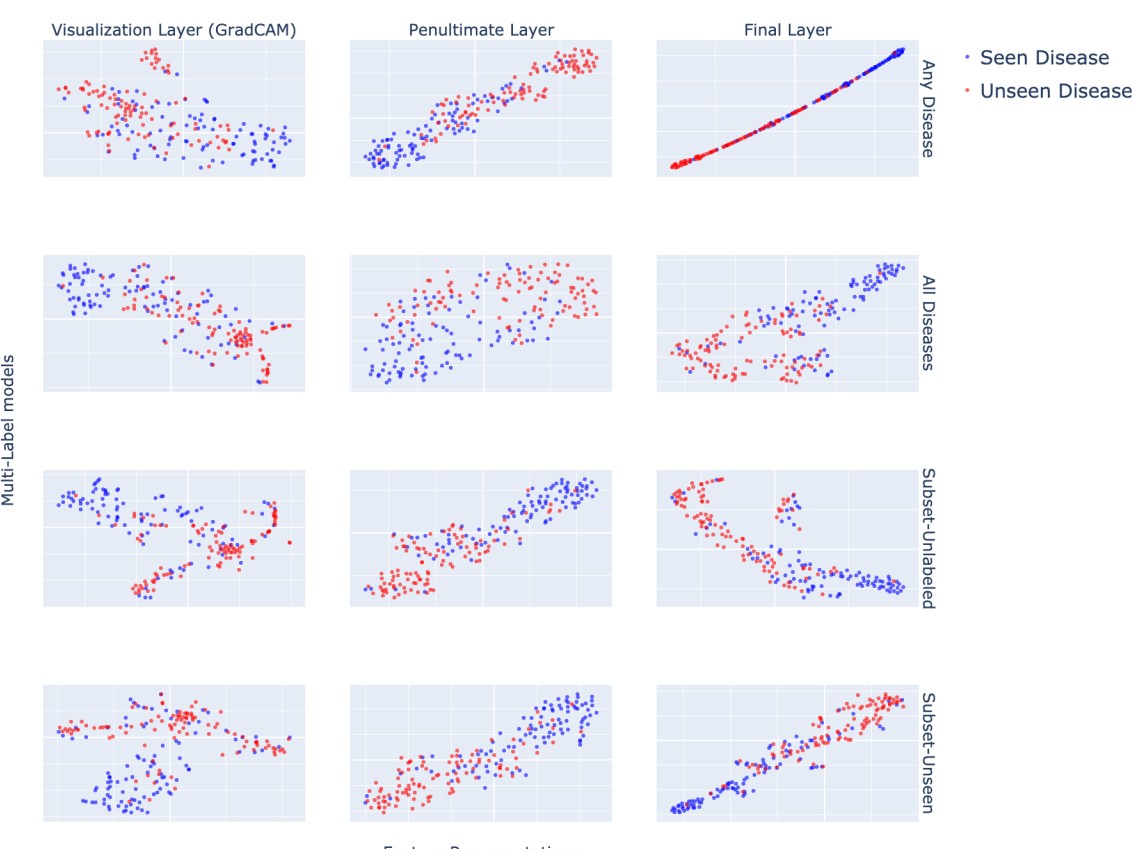

Figure 4: t-SNE plots of feature representations of each multi-label model

