# OpenReview forum: "Unseen Disease Detection for Deep Learning Interpretation of Chest X-rays"
_MIDL.io/2021/Conference — MIDL 2021_

### Official Review · AnonReviewer2 · 2021-03-07

**Confidence:** 4
**Preliminary Rating:** 3
**Recommendation:** Oral
**Final Rating:** 4

**Summary:**

The paper describes the experimental set-up and obtained results to answer three main questions: a) whether deep learning models trained on a subset of classes can detect the presence of any disease when tested on a dataset containing a larger set of classes, b) whether deep learning models trained on a set of diseases (seen diseases) can detect them when seen diseases co-occur with diseases outside the subset (unseen diseases), c) whether the representations learned by the deep learning models during training on seen diseases can be used to detect the presence of unseen diseases given a small labelled set of unseen diseases. To answer these questions, an existing dataset containing 10 classes was subdivided into two categories: seen diseases (4 classes), and unseen diseases (6 classes) based on the diseases’ prevalence in the dataset and four different multi-label models were trained.

**Strengths:**

1. The questions addressed in the paper are really important and relevant for the clinical deploy-ability of deep learning models. For a deep learning model to ensure reliability in real-world deployment, it should be able to identify unseen classes as anomalies so that recommendations for manual assessment could be made.
2. The experimental set up for partitioning the data to emulate the scenario - developing a deep learning model from prevalent diseases and deploying in the real-world where examples containing comparatively rarer diseases may also appear - is nicely done.
3. The questions raised in the introduction are addressed with sufficient experimental evidence.

**Weaknesses:**

1. One major weakness of the paper is stating statistical significance in the major portion of the results, without detailing the statistics used until the very end in the Appendix section. The details of statistical methods should have been described in the Methods section to allow the reader to interpret the results in the context of the statistical method used.
2. The authors state statistical significance while only the methods for assessing confidence interval and mean difference are described. A description of hypothesis testing is missing. Moreover, the reason for the choice of the specific statistical method is not explained. The results from multiple comparisons are reported without mentioning whether a correction for multiple comparisons was used or not.
3. The introduction gives an impression that some method on few-shot learning would be explored to see to what extent can the unseen classes be detected based on learning from a few labelled samples. However, not much focus has been given to this in the methods section.

**Deanonymize Review:**

no

**Final Rating Justification:**

The authors satisfactorily addressed the major issues raised in reviews during the rebuttal period.

**Justification Of The Preliminary Rating:**

The paper starts with raising really interesting and relevant questions. The experimental setup used for addressing these question is good. The results provide sufficient experimental evidence to the questions and are intuitive. However, only the surface has been touched for addressing the most important question (in my opinion): whether the representations learned by a deep learning model can be used for detecting unseen classes given a few labelled samples of unseen classes. Another weak point of the paper is describing majorly statistical results in the results section without detailing the statistical methods used in the main paper. It makes interpreting the results difficult.

**Paper Type:**

validation/application paper

**Special Issue:**

no

---

> ### Author Response · Authors · 2021-03-18
> **Thank you for the comments; suggestions well received and manuscript updated**
>
> We thank the reviewer for their encouraging comments. We have addressed their suggestions point by point and updated our manuscript accordingly.
>
> Comment:
> 1. One major weakness of the paper is stating statistical significance in the major portion of the results, without detailing the statistics used until the very end in the Appendix section. The details of statistical methods should have been described in the Methods section to allow the reader to interpret the results in the context of the statistical method used.
> 2. The authors state statistical significance while only the methods for assessing confidence interval and mean difference are described. A description of hypothesis testing is missing. Moreover, the reason for the choice of the specific statistical method is not explained. The results from multiple comparisons are reported without mentioning whether a correction for multiple comparisons was used or not.
>
> Response: For 1, we have now added additional details on our methods to establish statistical significance in Section 4 of the paper:
> “... We  calculate  p-values  from  the  confidence  interval  using  the method  described  in  (Altman  and  Bland,  2011)  with  a  threshold  of  0.05  for  hypothesis testing.”
>
> For 2, we have now added: “This method was chosen to evaluate whether 2 models were similar in performance with  respect  to  their  average  AUC  over  the  bootstrap  sample,  and  to  test  statistically significant performance differences in either direction using the 95% confidence intervals. We use the Benjamini-Hochberg method to correct for multiple hypothesis testing between various models.”
> We will also be updating the Tables in the appendix section with the p-values for each test.
>
> Comment: The introduction gives an impression that some method on few-shot learning would be explored to see to what extent can the unseen classes be detected based on learning from a few labelled samples. However, not much focus has been given to this in the methods section.
>
> Response: The reviewer suggests a good point for the paper. We evaluated whether feature representations extracted from the classification models can be useful for unseen disease detection. We conducted an initial exploration of few shot learning, focused on evaluation of feature extractions. Exploration of the optimal model for training the unseen disease classifiers using these features would be the important future research direction. We have updated the introduction and discussions to include the point.
> In introduction we added “We conduct an initial exploration of unseen disease detection methods, focused on evaluation of feature representations.”
> In discussions we added “We conduct an initial exploration of unseen disease detection methods, borrowing philosophy from few shot learning, while focusing on evaluating feature representations extracted from classification models.” and “Exploration of the optimal model for training the unseen disease classifiers using the feature representations evaluated in this project would be the important future research direction.”

---

### Official Review · ~Namkug_Kim1 · 2021-03-08

**Confidence:** 4
**Preliminary Rating:** 2
**Recommendation:** Poster

**Summary:**

This study evaluated the performance of deep learning models in the presence of diseases not labeled for or present during training. They modify the CheXpert dataset, consisting of 224,316 chest radiographs from 65,240 patients labeled for the presence of 14 observations (Irvin et al., 2019). However, the labels of this dataset was determined by NLP, which could lead to not accurate labels...

**Strengths:**

This study evaluated the performance of deep learning models in the presence of diseases not labeled for or present during training. it is unknown whether deep learning models for chest x-ray interpretation can maintain performance in presence of
diseases not seen during training, or whether they can detect the presence of such diseases.

**Weaknesses:**

They modify the CheXpert dataset, consisting of 224,316 chest radiographs from 65,240 patients labeled for the presence of 14 observations (Irvin et al., 2019). However, the labels of this dataset was determined by NLP, which could lead to not accurate labels...
in addition, the dataset has a limited number of diseases. Second, study is limited to an internal validation set without an external test set including different unseen diseases. study did not explore other training strategies for multi-label models that could mitigate the performance drop observed in our results with the All Diseases model compared to the Seen-Unlabeled model.

**Deanonymize Review:**

yes

**Detailed Comments:**

They modify the CheXpert dataset, consisting of 224,316 chest radiographs from 65,240 patients labeled for the presence of 14 observations (Irvin et al., 2019). However, the labels of this dataset was determined by NLP, which could lead to not accurate labels...
in addition, the dataset has a limited number of diseases. Second, study is limited to an internal validation set without an external test set including different unseen diseases. study did not explore other training strategies for multi-label models that could mitigate the performance drop observed in our results with the All Diseases model compared to the Seen-Unlabeled model.

**Justification Of The Preliminary Rating:**

They modify the CheXpert dataset, consisting of 224,316 chest radiographs from 65,240 patients labeled for the presence of 14 observations (Irvin et al., 2019). However, the labels of this dataset was determined by NLP, which could lead to not accurate labels...
in addition, the dataset has a limited number of diseases. Second, study is limited to an internal validation set without an external test set including different unseen diseases. study did not explore other training strategies for multi-label models that could mitigate the performance drop observed in our results with the All Diseases model compared to the Seen-Unlabeled model.

**Paper Type:**

validation/application paper

**Questions To Address In The Rebuttal:**

They modify the CheXpert dataset, consisting of 224,316 chest radiographs from 65,240 patients labeled for the presence of 14 observations (Irvin et al., 2019). However, the labels of this dataset was determined by NLP, which could lead to not accurate labels...
in addition, the dataset has a limited number of diseases. Second, study is limited to an internal validation set without an external test set including different unseen diseases. study did not explore other training strategies for multi-label models that could mitigate the performance drop observed in our results with the All Diseases model compared to the Seen-Unlabeled model.

**Special Issue:**

no

---

> ### Author Response · Authors · 2021-03-18
> **Comments received and addressed**
>
> We appreciate the reviewers’ comments. We address their concerns in detail in the comments, many of which were also detailed in the limitations section. On their point on limited number of diseases, this was addressed in the limitations and the dataset that we used, enriched because of NLP labeling, was one of the chest X-ray datasets with the highest variety of labels, second to PadChest. The setting of the experiments was adapted to the number of labels that were accessible, which is a fairly standard set-up.
>
> For their point on limited internal validation set and exploring other training strategies, the reviewer copied verbatim what was written in the limitations in our manuscript. For instance, our limitations section includes: “Third,  our  study  did  not  explore  other  training strategies for multi-label models that could mitigate the performance drop observed in our results with the All Diseases model compared to the Seen-Unlabeled model”. The reviewer wrote: “study did not explore other training strategies for multi-label models that could mitigate the performance drop observed in our results with the All Diseases model compared to the Seen-Unlabeled model.” Our “limitations” section includes “Second, our study is limited to an internal validation set without an external test set  including  different  unseen  disease“ and the reviewer wrote “Second, study is limited to an internal validation set without an external test set including different unseen diseases. “

---

### Official Review · AnonReviewer3 · 2021-03-09

**Confidence:** 5
**Preliminary Rating:** 2
**Final Rating:** 3

**Summary:**

The paper proposes different multi-label DL-trained models in different settings to find whether – 1) seen disease trained models can detect unseen diseases in test set (co-occurring with seen disease), 2) seen disease trained models can detect seen diseases in test set (co-occurring with unseen disease). The paper also explores how unseen diseases can be detected with or without explicit training. Results have been evaluated for CheXpert dataset with seen, unseen labels and no disease labels.

**Strengths:**

The paper presents a comprehensive comparative study conducted with 4 different multi-label models to evaluate the detection of unseen disease – if unseen diseases are present or absent in training, unseen diseases are present with or without labels. The clinical motivation of the  stems from the errors of labeling an unseen disease as “no disease” which can result in incorrect diagnosis. The paper also infers that the penultimate feature layer contains most discriminative features for unseen disease detection.

**Weaknesses:**

1) The paper lacks any state-of-the-art comparison study in any of the 4 settings. The authors use only two classifiers - logistic regression and random forest for multi-classification task.
2) The number of disease labels is limited.
3) Visualizations of penultimate and final layer features would have been insightful to illustrate the finding that the former ones are more significant in unseen disease detection.
4) How was statistical significance established? Though an attempt has been made to provide this information in the supplementary material, it is not clear
5) How robust is the ensembling?
6) How was the visualization map used for training?

**Deanonymize Review:**

no

**Final Rating Justification:**

The authors have addressed most of the critiques in the rebuttal and have provided an updated version. They propose addressing a few of the concerns as part of future work.

**Justification Of The Preliminary Rating:**

Though the paper addresses an important problem in the context of medical imaging and deep learning, the evaluation of the proposed methods is not exhaustive. Additionally, the generalizability of the model is not clear.

**Paper Type:**

both

**Questions To Address In The Rebuttal:**

See 'Weaknesses'

**Special Issue:**

no

---

> ### Author Response · Authors · 2021-03-18
> **Comments appreciated; manuscript updated according to suggestions**
>
> We thank the reviewer for their comments. We have addressed their suggestions point by point and updated our manuscript accordingly.
>
> Comment: The paper lacks any state-of-the-art comparison study in any of the 4 settings. The authors use only two classifiers - logistic regression and random forest for multi-classification task.
>
> Response: We agree that state-of-the-art unseen disease classifiers need to be developed for successful deployment of chest X-ray AI diagnosis models and is a future area of research. The goal of this work is to explore the quality of the feature representations extracted from the multilabel classification models in detection of unseen diseases, a first step towards building a state of the art unseen detection classifier. Future work would include building other classifiers that use the feature representations learned from our study.
>
> Comment: The number of disease labels is limited.
>
> Response: The number of disease labels were limited due to limitations in chest X-ray datasets, a common problem in medical imaging research and a motivation for our research. The diseases that we include are among the common diseases that chest X-ray datasets usually include (NIH Chest X-ray dataset, MIMIC-CXR dataset). Our work shows whether models trained with limited disease labels can still perform well when exposed to unseen labels and whether features from the models can be used for unseen disease detection.
>
> Comment: Visualizations of penultimate and final layer features would have been insightful to illustrate the finding that the former ones are more significant in unseen disease detection.
>
> Response: We have now added a method to visualize the learned representations used for the unseen scoring. In particular, we have added t-SNE plots for the feature representations that we used in the revised manuscript in Appendix B.3. From the revised manuscript:
> “To demonstrate the effectiveness of feature representations of multi-label models as inputsto unseen classifiers, we plot the 2D t-SNE (Van der Maaten and Hinton, 2008) clusters inFigure 4 of the feature representations for each multi-label model.  The t-SNE was run using a perplexity of 30 with 1000 iterations and a learning rate of 200.  The clusters are color coded with the disease subset label from the validation set (seen/unseen) that we describe in Section 3.1.”
>
> Comment: How was statistical significance established? Though an attempt has been made to provide this information in the supplementary material, it is not clear
>
> Response: We have amended the Statistical Analysis section (Section 4 in the paper) to read:
> “To determine statistical significance between 2 models, we use the 95% confidence intervals of the difference between bootstrap samples.  ...   We  calculate  p-values  from  the  confidence  interval  using  the method  described  in  (Altman  and  Bland,  2011)  with  a  threshold  of  0.05  for  hypothesis testing.  This method was chosen to evaluate whether 2 models were similar in performance with  respect  to  their  average  AUC  over  the  bootstrap  sample,  and  to  test  statistically significant performance differences in either direction using the 95% confidence intervals.We use the Benjamini-Hochberg method to correct for multiple hypothesis testing between various models.”
> We will also be updating the Tables in the appendix section with the p-values for each test.
>
> Comment: How robust is the ensembling?
>
> Response: The ensembling method we use follows the method used by CheXpert (Irvin et al, 2019) for the ensembling of their multi-label models. The interaction between the choices of the model (architecture, training procedure, ensembling strategy) and the detection of seen and unseen pathologies would be useful future work, but not the focus of our particular work.
>
> Comment: How was the visualization map used for training?
>
> Response: Output of the visualization map using GradCAM as a matrix is used directly to train the random forest or logistic regression classifier for unseen disease detection. We have updated the manuscript to make the step clear in the “unseen disease detection” section as “The output of the visualization map using GradCAM is used as a matrix directly in the following steps.”

---

### Official Review · AnonReviewer1 · 2021-03-09

**Confidence:** 3
**Preliminary Rating:** 3
**Recommendation:** Poster

**Summary:**

The authors evaluated the performance of deep learning models in the presence of diseases not labeled for or present during training. The unseen score was used as an output value to train classification models.
They showed the effectiveness of the safe clinical deployment of deep learning models trained on a non=-exhaustive set of disease classes.

**Strengths:**

The purpose of this paper is adequate to real environment situations in hospitals.
The consideration of using unseen scores is good in a clinical situations, which should be addressed when the application is adjusted in that situation.

**Weaknesses:**

1. Visualization method is needed to validate the effectiveness of using the unseen scoring. Just performance in terms of AUC is not enough to explain the strengths.
2. The lack of explanation for obtaining the unseen score specifically.
3. The lack of explanation for differences of random forest classifier and logistic regression classifier.

**Deanonymize Review:**

no

**Detailed Comments:**

P-value with the repeated test.

**Justification Of The Preliminary Rating:**

The use of the score for classifying the presence of the disease is basically a common issue in the general domain, but the concept of unseen data is additionally needed in clinical situations.
The author addressed this concept that should be focused on when applying it to this situation.


**Paper Type:**

validation/application paper

**Questions To Address In The Rebuttal:**

1. Visualization method is needed to validate the effectiveness of using the unseen scoring. Just performance in terms of AUC is not enough to explain the strengths. Can you add a figure or anything to show the effectiveness such as gradCAM?

2. The lack of explanation for obtaining the unseen score specifically.  Could you write more details on the process for the unseen scoring?

3. The lack of explanation for differences of random forest classifier and logistic regression classifier. Can you explain each purpose or the difference?

4. P-value with the repeated test. Can you measure the p-value to show significantly difference between methods?


**Special Issue:**

no

---

> ### Author Response · Authors · 2021-03-18
> **Thank you for the suggestions. We have addressed them and improved on manuscript.**
>
> We thank the reviewer for their encouraging comments. We have addressed their suggestions point by point and updated our manuscript accordingly.
>
> Comment: Visualization method is needed to validate the effectiveness of using the unseen scoring. Just performance in terms of AUC is not enough to explain the strengths. Can you add a figure or anything to show the effectiveness such as gradCAM?
>
> Response: We have now added a method to visualize the learned representations used for the unseen scoring. In particular, we have added t-SNE plots for the feature representations in Appendix B.3. From the revised manuscript:
> “To demonstrate the effectiveness of feature representations of multi-label models as inputs to unseen classifiers, we plot the 2D t-SNE (Van der Maaten and Hinton, 2008) clusters inFigure 4 of the feature representations for each multi-label model. The t-SNE was run using a perplexity of 30 with 1000 iterations and a learning rate of 200.  The clusters are color coded with the disease subset label from the validation set (seen/unseen) that we describe in Section 3.1.”
> The t-sne plot also demonstrates the non-linearity of the binary classification problem, which explains why the random forest classifier is able to perform better than the logistic regression classifier.
>
> Comment: The lack of explanation for obtaining the unseen score specifically. Could you write more details on the process for the unseen scoring?
>
> Response: We acknowledge that the unseen score was not explained in detail. We have expanded on the following paragraph to i the “unseen disease detection” section: “ Applying the four trained multi-label models to the validation set, we collect the outputs from the final prediction layer, the penultimate layer, and the visualization map (generated using the Grad-CAM method. The feature representations are extracted from running the validation set on the trained classification models. The three sets of outputs are then used to train a random forest classifier and a logistic regression classifier, with a binary outcome on whether the chest X-ray has an unseen disease or not, to produce an “unseen score'' using unseen disease labels on the validation set (shown in Figure 1C). Unseen scores are the output of the random forest classifier or the logistic regression classifier, and  a numeric number between 0 and 1, signifying how likely the chest X-ray image has an unseen disease.”
>
> Comment: The lack of explanation for differences of random forest classifier and logistic regression classifier. Can you explain each purpose or the difference?
>
> Response: We acknowledge that we did not explain why random forest classifier and logistic regression classifier were chosen. We have added the following explanation to the unseen disease detection section regarding the choice of including logistic regression and random forest classifier: “Logistic regression classifier is a commonly used standard for binary classification. Random forest classifiers, compared to logistic regression, are able to create nonlinear decision boundaries. We therefore used random forest in addition to the standard logistic regression as a strategy.”
>
> Comment: P-value with the repeated test. Can you measure the p-value to show significantly difference between methods?
>
> Response: We have amended the Statistical Analysis section (Section 4 in the paper) to read:
> “To determine statistical significance between 2 models, we use the 95% confidence intervals of the difference between bootstrap samples.  ...   We  calculate  p-values  from  the  confidence  interval  using  the method  described  in  (Altman  and  Bland,  2011)  with  a  threshold  of  0.05  for  hypothesis testing.  This method was chosen to evaluate whether 2 models were similar in performance with  respect  to  their  average  AUC  over  the  bootstrap  sample,  and  to  test  statistically significant performance differences in either direction using the 95% confidence intervals.We use the Benjamini-Hochberg method to correct for multiple hypothesis testing between various models.”
> We will also be updating the Tables in the appendix section with the p-values for each test.

---

### Meta-Review · Area_Chair1 · 2021-03-24

**Recommendation:** Accept (Poster)

**Metareview:**

The initial reviews were mixed, but the authors wrote a convincing rebuttal and made a revision of the manuscript, changing the initial rating of two reviewers to accept. Some reviewers did not provide a response to the rebuttal. Given the rebuttal of the authors and the changed opinion of two of the reviewers, I recommend acceptance for poster presentation.

**Paper Type:**

validation/application paper

---

### Decision · Program_Chairs · 2021-03-31

Accept